

# Choice of respiratory therapy for COVID-19 patients with acute hypoxemic respiratory failure: a retrospective case series study

Kazuki Sudo[1], Teiji Sawa[1,2], Kohsuke Kushimoto[1], Ryogo Yoshii[2], Kento Yuasa[1], Keita Inoue[2], Mao Kinoshita[1], Masaki Yamasaki[2] and Kunihiko Kooguchi[2]

[1] Department of Anesthesiology, Kyoto Prefectural University of Medicine, Kyoto, Japan
[2] Division of Intensive Care Unit, University Hospital, Kyoto Prefectural University of Medicine, Kyoto, Japan

Corresponding author
Teiji Sawa, anesth@koto.kpu-m.ac.jp

## ABSTRACT

**Background**. In the treatment of acute hypoxemic respiratory failure (AHRF) due to coronavirus 2019 (COVID-19), physicians choose respiratory management ranging from low-flow oxygen therapy to more invasive methods, depending on the severity of the patient's symptoms. Recently, the ratio of oxygen saturation (ROX) index has been proposed as a clinical indicator to support the decision for either high-flow nasal cannulation (HFNC) or mechanical ventilation (MV). However, the reported cut-off value of the ROX index ranges widely from 2.7 to 5.9. The objective of this study was to identify indices to achieve empirical physician decisions for MV initiation, providing insights to shorten the delay from HFNC to MV. We retrospectively analyzed the ROX index 6 hours after initiating HFNC and lung infiltration volume (LIV) calculated from chest computed tomography (CT) images in COVID-19 patients with AHRF.

**Methods**. We retrospectively analyzed the data for 59 COVID-19 patients with AHRF in our facility to determine the cut-off value of the ROX index for respiratory therapeutic decisions and the significance of radiological evaluation of pneumonia severity. The physicians chose either HFNC or MV, and the outcomes were retrospectively analyzed using the ROX index for initiating HFNC. LIV was calculated using chest CT images at admission.

**Results**. Among the 59 patients who required high-flow oxygen therapy with HFNC at admission, 24 were later transitioned to MV; the remaining 35 patients recovered. Four of the 24 patients in the MV group died, and the ROX index values of these patients were 9.8, 7.3, 5.4, and 3.0, respectively. These index values indicated that the ROX index of half of the patients who died was higher than the reported cut-off values of the ROX index, which range from 2.7–5.99. The cut-off value of the ROX index 6 hours after the start of HFNC, which was used to classify the management of HFNC or MV as a physician's clinical decision, was approximately 6.1. The LIV cut-off value on chest CT between HFNC and MV was 35.5%. Using both the ROX index and LIV, the cut-off classifying HFNC or MV was obtained using the equation, LIV = 4.26 × (ROX index) + 7.89. The area under the receiver operating characteristic curve, as an evaluation metric of the classification, improved to 0.94 with a sensitivity of 0.79 and specificity of 0.91 using both the ROX index and LIV.

**Conclusion**. Physicians' empirical decisions associated with the choice of respiratory therapy for HFNC oxygen therapy or MV can be supported by the combination of the ROX index and the LIV index calculated from chest CT images.

# INTRODUCTION

Coronavirus disease 2019 (COVID-19) is an emerging infectious disease currently causing a global pandemic. COVID-19 patients often present with mild symptoms; however, these may develop into more serious medical conditions, such as acute hypoxemic respiratory failure (AHRF) and septic shock, especially in older adults and patients with underlying illnesses. AHRF is a significant symptom in COVID-19 patients and requires the administration of high oxygen levels (*Attaway et al., 2021*; *Berlin, Gulick & Martinez, 2020*).

For mild AHRF associated with COVID-19, oxygen administration therapy using a nasal cannula or oxygen mask is the basic treatment strategy. However, for moderate or higher-severity AHRF, depending on the severity, high-flow nasal cannula (HFNC) oxygen therapy, mechanical ventilation (MV), or extracorporeal membrane oxygenation have been considered. Oxygen therapy with HFNC, which can provide a maximum oxygen flow of 60 L/min, has been used for COVID-19 patients who do not require MV (*Frat et al., 2015*; *Mellado-Artigas et al., 2021a*; *Roca et al., 2016a*). HFNC is more tolerable for patients than non-invasive ventilation (NIV) and MV (*Panadero et al., 2020*), and almost half of those who receive HFNC can be successfully weaned without the need for MV (*Calligaro et al., 2020*). However the use of HFNC in COVID-19 patients may delay the initiation of MV if respiratory failure worsens (*Kang et al., 2015*). The failure of HFNC has been associated with increased mortality compared with the failure of NIV and MV alone (*Miller et al., 2022*). Therefore, when treating COVID-19 patients with AHRF, it is critical to appropriately evaluate whether to continue treatment with HFNC or to initiate MV.

Since the beginning of the COVID-19 pandemic, which began in Japan in April 2020, clinicians have considered the risk factors that influence the course of the disease when choosing respiratory therapy. Among the physiological parameters, the ratio of oxygen saturation (ROX) index ($SpO_2 \times$ respiratory rate$^{-1} \times F_iO_2^{-1}$, which is the combination of percutaneous blood oxygen saturation, respiratory rate, and inspired oxygen concentration, respectively) is a useful indicator to evaluate the severity of AHRF in COVID-19 patients (*Roca et al., 2019*; *Roca et al., 2016b*). Basically, lower ROX index values are associated with higher AHRF severity. The cut-off value of the ROX index is a proposed criterion for discontinuing HFNC and initiating NIV or tracheal intubation for MV (*Ferrer et al., 2021*; *Hu et al., 2020*; *Vega et al., 2022*). However, although the ROX index could be a potential marker to identify patients with a higher risk of HFNC failure, the prediction efficiency is moderate, and the optimal cut-off value and the acquisition time of the ROX index

continue to be discussed (*Junhai et al., 2022*). In fact, in our facility, as shown in this article, some patients died even if the ROX index was higher than the cut-off values reported by others. Conversely, other patients were saved using HFNC even if the ROX index was much lower than the cut-off values.

Since the start of the COVID-19 pandemic, the role of chest computed tomography (CT) in the management of COVID-19 patients has evolved in terms of the indications in the acute phase and the prediction of pathological conditions in the subacute phase (*Komurcuoglu et al., 2022*; *Lyu et al., 2020*; *Machnicki et al., 2021*; *Sayeed et al., 2021*). COVID-19 pneumonia is characterized by extensive infiltration shadows in the lungs on chest CT images. Thus, chest CT in COVID-19 patients has provided radiological information of the severity of pneumonia. Additionally, clinicians can make judgments about treatment options by assessing the oxygenation-associated physiological parameters and other parameters associated with medical image analysis, such as the evaluation of pneumonia severity. In this study, in COVID-19 patients with AHRF, we retrospectively analyzed the ROX index 6 h after the initiation of HFNC and other parameters, including lung infiltration volume (LIV) calculated from chest CT images. Using the cut-off of the ROX index to determine HFNC oxygen therapy and MV initiation, we devised an objective index to achieve empirical physician decisions for initiating MV, providing insights to shorten the delay from HFNC to MV.

## MATERIALS & METHODS

### Research ethics

This study was conducted in accordance with the guidelines of the Declaration of Helsinki as a retrospective observational study accompanying the Kyoto Prefectural University of Medicine (KPUM) COVID-19 Registry Study (ERB-C-1810, approved by the Institutional Review Board of KPUM on 3 September 3, 2020). This study evaluated only adult COVID-19 patients. Written informed consent was obtained from all subjects and/or their legal guardian(s). The informed consent form consists of two parts: the information sheet that outlines the nature of the project, activities involved, timeframe, expectations of both the researcher and participants, data collection and storage methods, how the data will be used, and whether there will be any risk or benefit to the participants, and the consent certificate on which investigators obtained a signature from the participant. All methods were performed in accordance with the relevant guidelines and regulations.

### Target patients and the choices of respiratory therapies

The KPUM Hospital is a nationally accredited first-class infectious disease-designated hospital in Kyoto Prefecture, Japan. This hospital has been performing inpatient treatment for COVID-19 patients with severe respiratory failure, mainly *via* referral requests from other medical institutions in Kyoto Prefecture to control centers in Kyoto Prefecture. From April 2020 to September 2021, 188 patients diagnosed as COVID-19-positive were hospitalized (Fig. 1). Of these, 112 were mildly ill patients who did not require advanced oxygen therapy. Of the 76 patients who required high-flow oxygen therapy, after excluding 14 patients who had already been hospitalized and were receiving MV and three patients

who did not receive MV because of palliative care, 59 patients who started HFNC therapy immediately after admission were the subjects of this study. As a result of their choice of respiratory management, patients who were indicated for MV and managed with MV were classified into the MV group, and those who were successfully managed with HFNC were classified into the HFNC group. HFNC therapy was started using Optiflow (Fisher & Paykel Healthcare, Auckland, New Zealand) to maintain a respiratory rate of less than 30 breaths per minute by adjusting oxygen flow and oxygen concentration. The indication for MV was empirically determined by the attending physicians in charge of the patient with reference to the patient's age, comorbidities, oxygenation assessment, and chest CT images. The major criteria were: hypoxemic respiratory failure with $SpO_2$ <90% or a ratio of the partial pressure of arterial oxygen to $FiO_2$ of <200 despite receiving the maximal $FiO_2$ possible with HFNC; hypercapnic respiratory failure accompanied by blood pH <7.3; respiratory rate >30 breaths per minute; and hypotension (systolic blood pressure <90 mmHg) despite catecholamine and/or fluid administration. The following additional data were collected at admission: age, gender, weight, height, body mass index (BMI), comorbidities (hypertension, diabetes, lung disease, heart disease, cerebrovascular disease), blood clinical laboratory data, pneumonia severity index (*Fine et al., 1997*), and Charlson comorbidity index (*Charlson et al., 1987*).

### ROX index
The oxygen flow rate was adjusted according to the patient's body condition, and the concentration was adjusted so that $SpO_2$ was maintained at $\geq$ 95% at rest. The ROX index was then calculated approximately 6 h after admission.

### Chest CT analysis
All patients underwent CT before transfer to our hospital or immediately after admission. 3D Slicer software (ver.4.11, https://www.slicer.org/) was used to calculate the ratio of LIV by chest CT image analysis (*Balbi et al., 2021*; *Cattabriga et al., 2020*; *Digumarthy et al., 2019*). According to each Hounsfield unit value, the segmented lung images were color-coded using one mm-volume reconstructions. The LIVs were calculated and expressed as percentages. Chest CT images of the HFNC and MV groups were analyzed using 3D Slicer to determine the volume of the normal lung range and the ratio of the LIV (*Lanza et al., 2020*).

### Kernel density graph
Open-source Python (ver. 3.8; https://www.python.org) with the Seaborn (https://seaborn.pydata.org) library was used for graphing.

### Estimation of the cut-off points based on the physicians' decisions
If the choice of respiratory therapy by the attending physician was considered a problem regarding the accuracy of the two-group classification, the choice was replaced with an analysis of sensitivity and specificity by plotting the percentages of MV cases with a ROX index >ROX cut-off (true positive ratio, or sensitivity) curve and the percentages of HFNC cases with a ROX index <ROX cut-off (true negative ratio, or specificity) curve *vs*

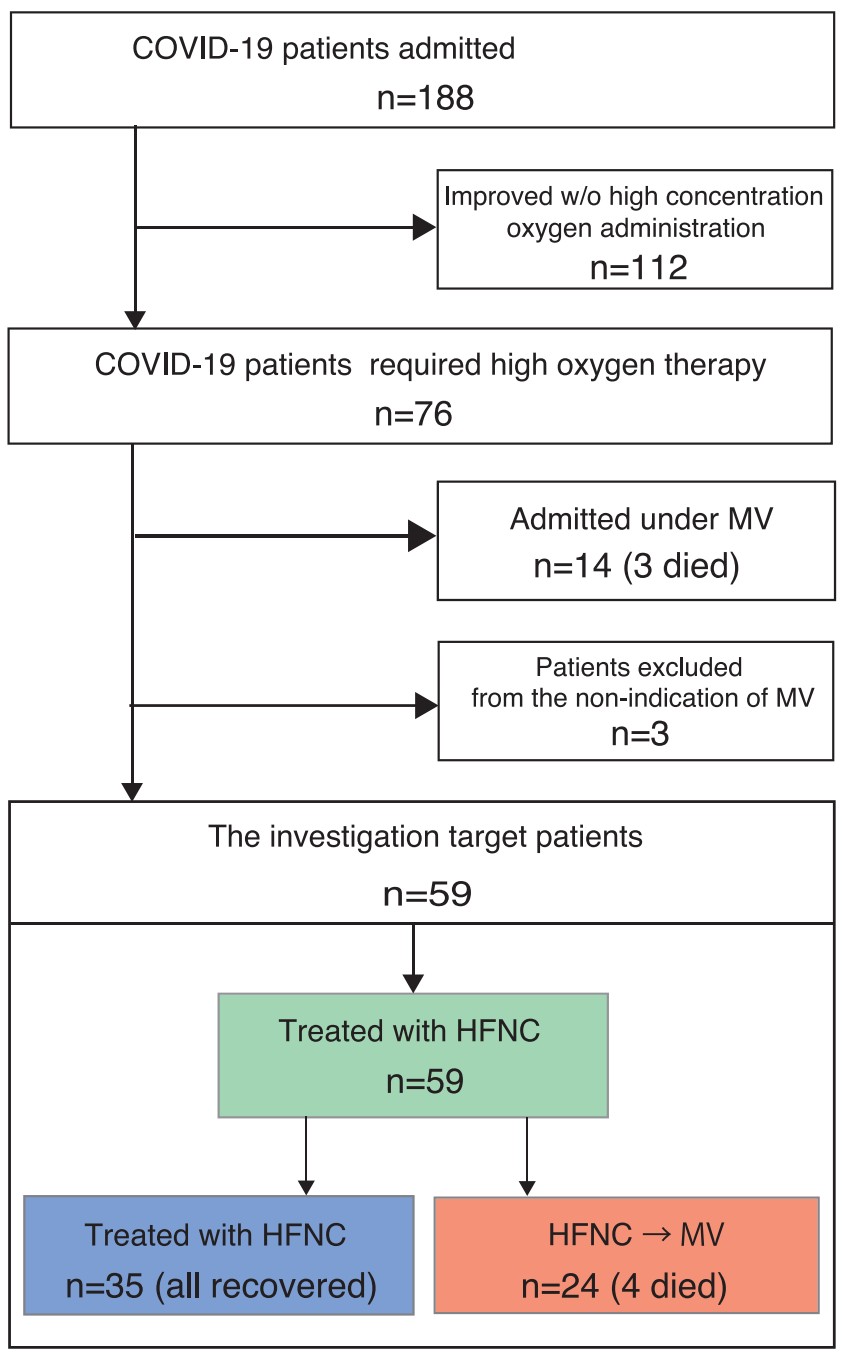

**Figure 1 Patient flowchart.** One hundred eighty-eight patients were referred to the University Hospital of Kyoto Prefectural University of Medicine from April 2020 to September 2021; 122 patients were mildly ill and did not require high-flow oxygen therapy. Of the 76 severe COVID-19 patients who required high-flow oxygen therapy, 59 patients received HFNC therapy after admission after excluding 3 patients who did not receive MV because of palliative care and 14 patients who had already been hospitalized under MV. Thirty-five patients completed treatment with HFNC and 24 were intubated for management with MV. HFNC, high-flow nasal cannulation; MV, mechanical ventilation; HFNC → MV, cases transitioned from HFNC to MV.

cut-off values to identify a threshold for MV use by physicians. The cut-off value that the attending physicians subconsciously selected was considered the crossover point where the proportion of false negatives (HFNC cases with a ROX index ≤ cut-off) equaled the proportion of false positives (MV cases with a ROX index >cut-off) (*Greiner, Sohr & Göbel, 1995*; *Yoshida et al., 2017*).

We assumed that LIV directly captured the severity of lung injury; therefore, the crossover point value was obtained by superimposing the LIV cut-off value (= 35.5) for false positives and false negatives over those for the cut-off of the ROX index alone and the cut-off that varied with the ROX index. In the ROX alone judgment, it was possible to explore the effect of LIV on effects that were classified as false. Accordingly, we calculated the proportion of HFNC cases with LIV ≤ 35.5 and ROX index ≤ ROX cut-off, and the percentage of MV cases with LIV >35.5 and ROX index >ROX cut-off.

## Multiple binomial logistic regression analysis

R programming language (*R Core Team, 2022*) was used for the multiple binomial logistic regression analysis (MLRA) to calculate the decision border (cut-off) for the classification of HFNC or MV groups. In R function *glm* (response~epredictor, family = binomial (link = "logit"), dependent variable data), using a series of attributes for each patient as the dependent variable data, we described the choice of respiratory therapy, which is a predictor variable, in MV (1), HFNC (0), as binary. For example, in the classification of HFNC or MV groups using the ROX index and/or LIV as the dependent variables, logit() can be expressed as

(1) ROX index alone: $\text{logit}(p_i) = \text{SCORE} = \ln(p_i/(1 - p_i)) = \beta_0 + \beta_1 \times [\text{ROX index}]$, where $p_i$ is the probability of the event and $\beta_0$ is the intercept from the linear regression equation; $\beta_1$ is a parameter. In this case, the decision borderline for the classification is described as $[\text{ROX index}] = -\beta_0/\beta_1$, where $p_i = 0.5$, and SCORE = 0 is the cut-off value.

(2) LIV alone: $\text{logit}(p_i) = \text{SCORE} = \ln(p_i/(1 - p_i)) = \beta_0 + \beta_2 \times [\text{LIV}]$, where $p_i$ is the probability of the event, and $\beta_0$ is the intercept from the linear regression equation; $\beta_2$ is a parameter. In this case, the decision borderline for the classification is described as $[\text{LIV}] = -\beta_0/\beta_2$, where $p_i = 0.5$ and SCORE = 0 is the cut-off value.

(3) ROX index and LIV: $\text{logit}(p_i) = \text{SCORE} = \ln(p_i/(1 - p_i)) = \beta_0 + \beta_1 \times [\text{ROX index}] + \beta_2 \times [\text{LIV}]$, where $p_i$ isthe probability of the event and $\beta_0$ is the intercept from the linear regression equation; $\beta_1$ and $\beta_2$ are parameters. In this case, the decision borderline for the classification is described as $[\text{LIV}] = -\beta_1/\beta_2 \times [\text{ROX index}] - \beta_0/\beta_2$, where $p_i = 0.5$ and SCORE = 0 is the cut-off value.

## Receiver operating characteristic (ROC) analysis

Under the above-described definitions of the binomial classifications associated with the HFNC or MV groups and three different cut-off lines (ROX index = 6.1, LIV = 35.5, and LIV = 4.26× (ROX index) + 7.89), we performed a ROC analysis to evaluate the reliability of classification by each cut-off value. In addition to the sensitivity: TP/(TP + FN), the specificity: TN/(FP + TN), ROC curve, and area under the summary ROC curve

(AUC), we calculated accuracy: (TP + TN)/[Total ($n = 59$)]; positive likelihood ratio (PLR): sensitivity/(1 − specificity); negative likelihood ratio (NLR): (1 − sensitivity)/specificity; and diagnostic odds ratio (DOR): PLR/NLR.

## Other statistical analyses

SPSS (ver. 27; IBM Corp., Armonk, NY, USA) was used for the unpaired $t$-test to compare group means, and Microsoft Excel was used for the $\chi^2$ test for comparisons between groups. Data are shown as mean ± standard deviation (sd).

## RESULTS

### Patients' therapeutic backgrounds

The first wave of the COVID-19 pandemic started in April 2020 in Kyoto, and five pandemic waves occurred by September 2021. From December 2020, more active use of HFNC was promoted at our facility, and as a result, the number of patients who underwent HFNC or MV management increased gradually until September 2021, the end of the study period (Fig. S1). During the 18-month study period, among the hospitalized patients, three were under HFNC therapy and were not candidates for MV therapy; these patients were excluded from this study per the hospital's code of ethics. Fourteen patients had been mechanically ventilated under tracheal intubation by the time they were transferred to our hospital. The remaining 59 patients were the target of further analysis in this study. Of these patients, within 2.8 ± 3.6 days, 24 were indicated for MV and were changed to MV management under tracheal intubation. We compared the primary data of the 35 patients who were successfully treated with HFNC (HFNC group) and the 24 patients who required MV (MV group). Note that the number of ventilators available at our hospital could have been the upper limit of the number of patients in the MV group. However, in the study period during the COVID-19 pandemic, the number of ventilated patients never reached the upper limit of the number of ventilators available at our hospital. No statistically significant difference was detected for gender, age, body weight, height, and BMI (Table 1). Regarding the presence or absence of underlying disease, no statistically significant difference was detected, except for chronic kidney disease, which was higher in the MV group (Table S1). The primary treatment comprised antiviral drugs, such as favipiravir or remdesivir, anti-immunotherapy, mainly with dexamethasone, and anticoagulant therapy with heparin. There was no significant difference in drug therapy between the two groups (Table S2). A multidisciplinary conference was held by the attending physician and infectious disease specialist, infectious disease control team, and intensive care specialist, and baricitinib, tocilizumab, and steroid pulse therapy were given as additional anti-immunotherapies when needed. No patients were treated with monoclonal antibodies and none were vaccinated.

The patients' blood laboratory test data showed significantly higher lactate dehydrogenase concentrations at admission in the MV group compared with the HFNC group (Table 1). The mean ROX index value in the HFNC group was significantly higher compared with the MV group (Table 1). Regarding the analysis of chest CT images by 3D Slicer, the LIVs and their proportions were significantly higher in the MV group compared

**Table 1  Major characteristics of HFNC and MV groups.**

| Characteristics | HFNC | MV | *p*-value |
|---|---|---|---|
| *n* | 35 | 24 | |
| Female/male, n/n | 9/26 | 6/18 | 0.951 |
| Age (years old) | 61.1 ± 12.3 (43–84) | 58.0 ± 14.5 (36–81) | 0.403 |
| Body weight (kg) | 69.4 ± 16.3 (41.7–106.5) | 73.4 ± 17.5 (44.6–127) | 0.378 |
| Height (cm) | 166.4 ± 8.7 (149–184) | 166.4 ± 10.7 (137–185) | 0.988 |
| Body mass index (kg/m$^2$) | 24.8 ± 4.4 (18.5–34.8) | 26.6 ± 6.5 (17.6–48.1) | 0.253 |
| Period from onset to admission to our hospital (days) | 9.7 ± 2.7 (4–17) | 7.5 ± 3.1 (2–16) | 0.008[*] |
| Period from onset to the introduction of HFNC (days) | 9.8 ± 2.7 (6–17) | 7.5 ± 2.5 (3–13) | 0.001[*] |
| Laboratory data (admission) | | | |
| White blood cells (/μL) | 8257 ± 5377 (1800–25600) | 7775 ± 4259 (1500–16900) | 0.703 |
| Creatinine (mg/dL) | 1.3 ± 2.0 (0.44–10.63) | 1.7 ± 2.2 (0.37–10.55) | 0.233 |
| C-reactive protein (mg/L) | 9.3 ± 7.3 (0.55–32.16) | 10.6 ± 7.7 (1.2–31.2) | 0.508 |
| Lactate dehydrogenase (U/L) | 397.0 ± 72.1 (226–598) | 557.5 ± 237.8 (122–1086) | 0.004[*] |
| D-dimer (mg/L) | 2.9 ± 7.2 (0.3–36.0) | 3.3 ± 5.7 (0.5–21.7) | 0.820 |
| Indices for organ damage | | | |
| Pneumonia severity index | 86.8 ± 27.8 (43–139) | 102.8 ± 51.8 (29–245) | 0.175 |
| Charlson comorbidity index | 1.7 ± 2.0 (0–10) | 2.0 ± 2.0 (0–8) | 0.612 |
| Lung analysis | | | |
| Lung infiltration volume (mL) | 972.2 ± 321.7 (518–1845) | 1340 ± 482 (438–2319) | 0.002[*] |
| Lung infiltration volume (%) | 26.7 ± 7.8 (9.8–38.4) | 41.9 ± 11.7 (15.5–72.2) | <0.001[*] |
| ROX index | 7.7 ± 2.4 (4.4–17.1) | 5.4 ± 1.8 (2.8–9.8) | <0.001[*] |

**Notes.**

The data are shown as mean ±sd (range).

[*] *p* < 0.05, statistically significant difference between HFNC and MV.

HFNC, high-flow nasal cannula; MV, mechanical ventilation; ROX index, ratio of oxygen saturation index.

with the HFNC group (Table 1) Fig. 2 and Video Clip A–F show the analysis of the chest CT images of six cases with different pneumonia severity according to 3D Slicer. The period from onset to admission to our hospital and from onset to intervention with HFNC were significantly longer in the HFNC group than those in the MV group (Table 1).

As a clinical outcome in both groups, the length of hospital stay was significantly longer in the MV group compared with the HFNC group (Table S3). Patients in the HFNC group were intubated and transferred to the MV group if their respiratory status deteriorated. Therefore, no deaths occurred in the HFNC group; however, four patients died in the MV group (Table S3). In the HFNC group, HFNC was performed for an average of 7.1 ± 10.3 (range: 1–62) days. In the MV group, the average period from HFNC to MV was 2.8 ± 3.6 (range: 0–16) days, and this was followed by 15.2 ± 23.6 (range 2–97) days of MV (Table S3).

## The various cut-off levels of the ROX index and the clinical outcomes

As stated, no patients died among the 35 patients who received HFNC until their recovery (because patients who were initially receiving HFNC but who were later intubated owing to worsening respiratory status were subsequently assigned to the MV group). In contrast, four of the 24 patients in the MV group died. The ROX index values of these patients were

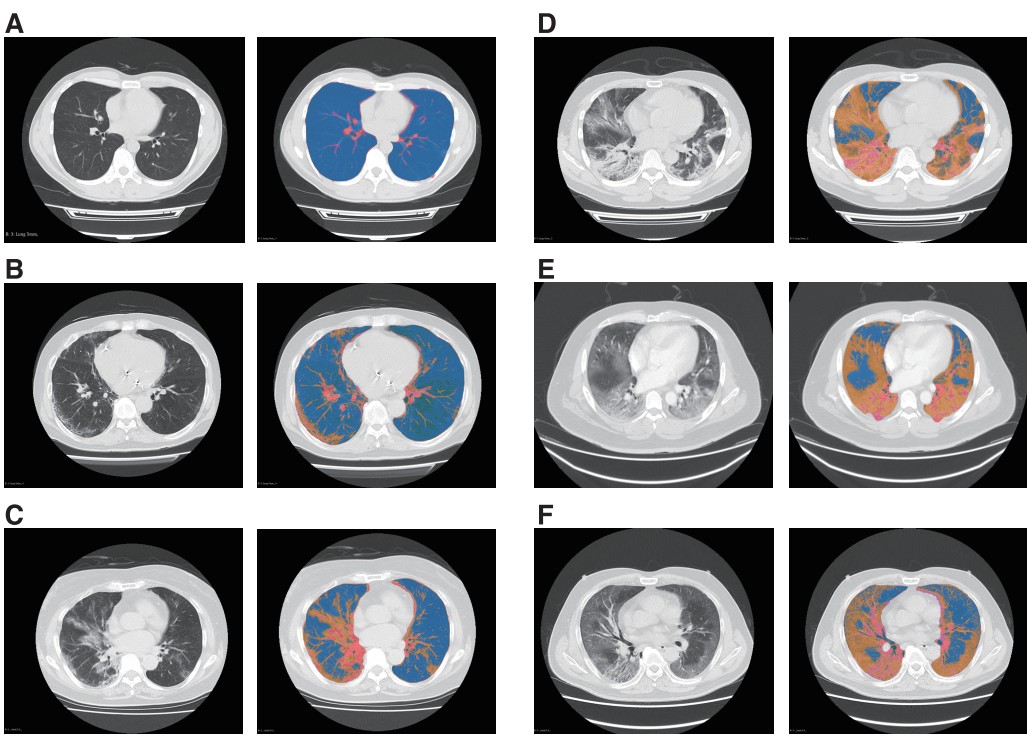

**Figure 2 Chest CT images.** Chest CT settings were as follows: voltage, 120 kV; tube current, 266 mA; slice thickness, 5.00 mm; window width, 1500 Hounsfield units (HU); window level, −600 HU. According to the different HU intervals, lung volumes were segmented and extracted as follows: emphysema (density between −1,050 HU and −950 HU), normal lung ventilation (density between −949 HU and −750 HU), infiltration shadow (density between −749 HU and −400 HU), collapsed lung (density between −399 HU and 0 HU), and blood vessels and soft tissue (density between 1 HU and 1,000 HU). Chest CT images were read with 3D Slicer software and classified into normal infiltration, blood vessels, and emphysema according to the volume of 1 mm³ unit of CT concentration. (A). Findings in a patient who did not require oxygen administration. Most findings are normal. (B) Findings in a patient who was successfully treated with low-flow oxygen therapy. A slight infiltration shadow is seen dorsally. (C) Findings in a patient who was successfully treated with HFNC. Infiltration shadows are seen extensively dorsally. This patient was effectively treated in the prone position. (D) A patient treated with HFNC for several days who failed HFNC and was transitioned to MV. The patient had diffuse ventral shadows on imaging. Therapy in the prone position was not effective in this patient. (E) Findings in a patient who was treated with HFNC but was transitioned to MV on the same day. Extensive infiltration shadows are noted. (F) Findings in a patient treated with HFNC for several days and subsequently transitioned to MV. Infiltration shadows are observed in most of the lung fields. This patient was unable to maintain oxygenation after initiation of MV and required extracorporeal membrane oxygenation. CT, computed tomography; HFNC, high-flow nasal cannulation; MV, mechanical ventilation.

9.8, 7.3, 5.4, and 3.0, respectively, suggesting that the ROX index of half of the patients who died was higher than the reported cut-off values of the ROX index, which range from 2.7 to 5.99 (*Prakash et al., 2021*).

The MV group comprised seven patients with ROX index values ≥ 6 and five had LIV values ≥ 35.5%, indicating severe lung injury. Conversely, two of the 34 survivors in the HFNC group had a ROX index of ≤ 5. Therefore, the attending physicians selected respiratory therapy (HFNC or MV) without being bound only by the ROX index. When

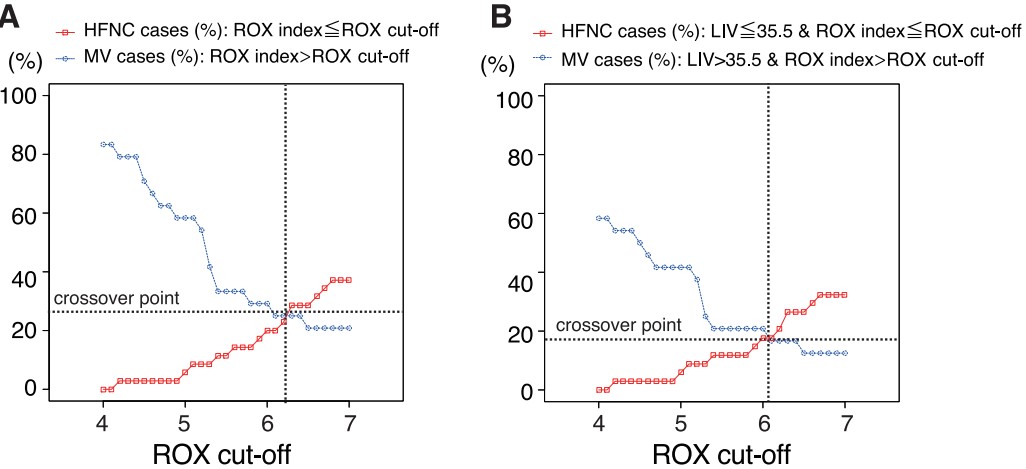

**Figure 3  The relationships between the cut-off values of the ROX index and the respiratory therapeutic choice (HFNC or MV).** (A) The percentages of HFNC cases with a ROX index ≤ ROX cut-off and MV cases with a ROX index > ROX cut-off. (B) The percentage of HFNC cases with an LIV ≤ 35.5 and ROX index ≤ ROX cut-off, and the percentage of MV cases with LIV > 35.5 and ROX index > ROX cut-off. HFNC, high-flow nasal cannula oxygen therapy; LIV, lung infiltration volume; MV, mechanical ventilation; ROX index, ratio of oxygen saturation index.

the cut-off value of the ROX index varied from four to seven in increments of 0.1, we calculated the proportion of HFNC patients whose ROX index was ≤ the cut-off value and the proportion of MV patients whose ROX index was >the cut-off value. The proportion of HFNC patients with ROX index values ≤ the cut-off value and the proportion of MV patients with ROX index values >the cut-off value crossed over at 25%, where the cut-off value of the ROX index was approximately 6.2 (Fig. 3A).

Next, we calculated the percentage of HFNC patients with LIV values ≤ 35.5 (we explained this LIV cut-off value in the next section) and ROX index ≤ the cut-off value and the percentage of MV patients with LIV values >35.5 and ROX index >the cut-off value. The percentage lines of both HFNC and MV patients crossed over at 17%, where the cut-off value of the ROX index was approximately 6.1 (Fig. 3B). These results mean that the judging criteria for the cut-off value of the ROX index by the attending physician was approximately 6.1–6.2, which is slightly higher than the reported cut-off value of the ROX index (2.7–5.99) (*Prakash et al., 2021*). Thus, by adding LIV = 35.5 as a cut-off to ROX index = 6.1, the crossover point, at which the proportion of false positives in the HFNC group matched the proportion of false negatives in the MV group, decreased from 25% to 17% under the ROX index cut-off of about 6.1. This finding suggests that the addition of LIV to the treatment decision contributed to reducing false positives in HFNC cases (HFNC cases with ROX index ≤ ROX cut-off) and false negatives in MV cases (MV cases with ROX index >ROX cut-off), and the accuracy (1 − (false positives + false negatives)/total cases) increased in the sensitivity and specificity analyses.

**Table 2  Covariate results used for multiple logistic analysis.**

| Covariates | Odds ratio | 95% CI | p-value |
|---|---|---|---|
| Laboratory data (admission) | | | |
| Lactate dehydrogenase | 1.01 | 1.00–1.02 | 0.09 |
| Period from onset to admission to our hospital (days) | 0.67 | 0.42–1.08 | 0.10 |
| Period from onset to the introduction of HFNC (days) | 0.89 | 0.54–1.46 | 0.64 |
| Lung analysis | | | |
| Lung infiltration volume (%) | 1.25 | 1.06–1.46 | 0.008[*] |
| ROX index | 0.32 | 0.13–0.77 | 0.012[*] |

Notes.

CI, confidence interval; HFNC, high-flow nasal cannula; ROX index, ratio of oxygen saturation index.

## MLRA of the indications for HFNC and MV

MLRA was performed using the five factors involved in the decision to initiate MV management: the period from onset to admission to our hospital, the period from onset to the initiation of HFNC, laboratory examination data (lactate dehydrogenase concentration), a lung injury parameter (LIV) from chest CT imaging, and the ROX index (Table 2). Note that we did not include characteristics related to history and underlying diseases for the MLRA because these diagnostic criteria are ambiguous (Table S1). Covariates with p-values $\geq 0.05$ were excluded from the regression analysis (Table 2). As a result, the results for the ROX index (odds ratio, 0.32; 95% CI [0.13–0.77]; $p = 0.012$) and LIV on chest CT images (odds ratio, 1.25; 95% CI [1.06–1.46]; $p = 0.008$) were significant. Note that the pairs plot shows significantly different distributions for the ROX index and LIV when the patients were divided into two groups (MV group and HFNC group) (Fig. S2). Next, MLRA was repeated using only the ROX index and LIV. Optimal cut-off values for the ROX index and LIV were then determined for the two management groups (38 patients who underwent MV and 35 patients who were treated with HFNC alone). As a result, when using the ROX index alone as a cut-off value, the boundary score (SCORE) for classifying patients selected for HFNC or MV was calculated as SCORE = $\ln(p_i/(1 - p_i)) = 4.21 - 0.69 \times$ [ROX index], and [ROX index] = 6.1 when $p_i = 0.5$, indicating SCORE = 0. Therefore, the cut-off value of the ROX index was 6.1. When using LIV alone as a cut-off value, the boundary score (SCORE) for classifying patients selected for HFNC or MV was calculated as SCORE = $\ln(p_i/(1 - p_i)) = -8.09 + 0.23 \times$ [LIV], and [LIV] = 35.5 when $p_i = 0.5$, indicating SCORE = 0. Therefore, the cut-off value of LIV was 35.5%. Finally, when using both the ROX index and LIV as cut-off values, the boundary score (SCORE) for classifying patients selected for HFNC or MV was calculated as SCORE = $\ln(p_i/(1 - p_i)) = -1.50 - 0.81 \times$ [ROX index] + 0.19 × [LIV]. Therefore, the decision borderline was calculated as [LIV] = 4.26 × [ROX index] + 7.89 when $p_i = 0.5$, indicating SCORE = 0.

We plotted all 59 patients by ROX index and LIV values with color codes demonstrating HFNC or MV (cases of transition from HFNC to MV), and drew the distribution density as a kernel density estimation (KDE) plot (Fig. 4A). The KDE plot indicated that higher patient density was associated with more concentrated patient distribution. Next, the KDE

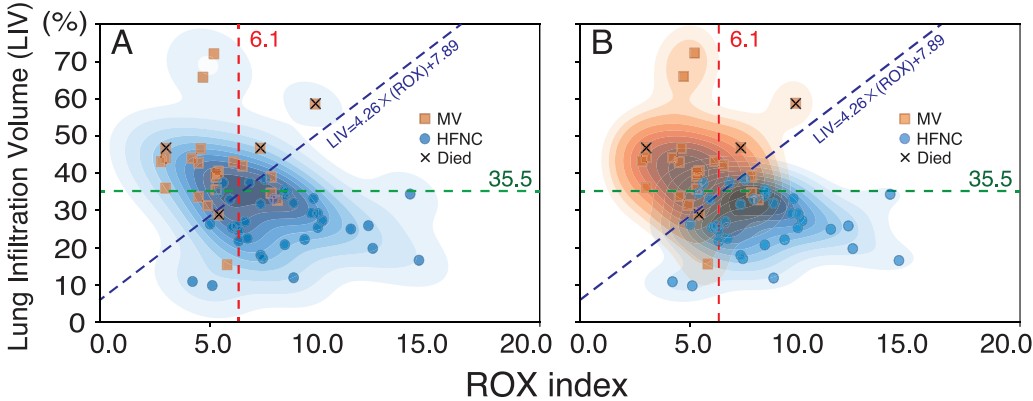

**Figure 4 Kernel density estimation of patient distribution, HFNC, and ventilator management.** The decision boundaries (cut-off lines) to classify the HFNC and MV groups were calculated by multiple logistic regression analysis using the ROX index and/or LIV. The cut-off value to classify the HFNC and MV groups by ROX index was 6.1 and that by LIV was 35.5. The cut-off line to classify the two groups using the ROX index and LIV was calculated as (LIV) = 4.51 × (ROX index) + 1.75. A. Kernel density plot using all 59 patients' data. B. Kernel density plots for the MV and HFNC groups. HFNC, high-flow nasal cannula; MV, mechanical ventilation; HFNC → MV, cases transitioned from HFNC to MV; ROX index, ratio of oxygen saturation index.

plot was drawn separately for the HFNC and MV groups, namely 35 patients who were treated with HFNC alone and 24 patients who underwent MV (Fig. 4B). As shown in Fig. 4A, the highest density in the center of the KDE plot indicates the crossing point of three decision borderlines, and, as shown in Fig. 4B, the KDE plots, which were drawn separately for the HFNC and MV groups, overlapped noticeably near the crossing point of the three decision borderlines. These findings suggested that the physicians' classification of HFNC or MV was problematic for patients whose indices were close to the statistically calculated cut-off values in this study. Considering that no patients in the HFNC group died, the association between a ROX index <6.1 and the prediction of mortality outcome with MV treatment must be related to the validity of the physicians' clinical decisions, which is a limitation of this study.

## Cut-off by ROX index and/or LIV for the classification of HFNC or MV

With 6.1 as the cut-off for the ROX index, 18 (75.0%) of the 24 patients managed with MV were classified as the severe group, and 32 (80.0%) of the 35 patients managed with HFNC were classified as the mild group (Table. S4). In contrast, when the LIV cut-off was 35.5%, 18 (75.0%) of the 24 patients managed with MV were classified as the severe group, and 31 (88.6%) of the 35 patients managed with HFNC were classified as the mild group (Table. S4). As shown in Fig. 3, compared with the vertical cut-off line with a ROX index of 6.1 alone, the cut-off line by MLRA SCORE LIV = 4.26 × (ROX index) + 7.89 was tilted in the positive direction of the ROX index and the LIV axes. When using the SCORE cut-off, 19 (79.2%) of the 24 the patients managed with MV were classified as the severe group, and 32 (91.4%) of the 35 patients managed with HFNC were classified as the mild group

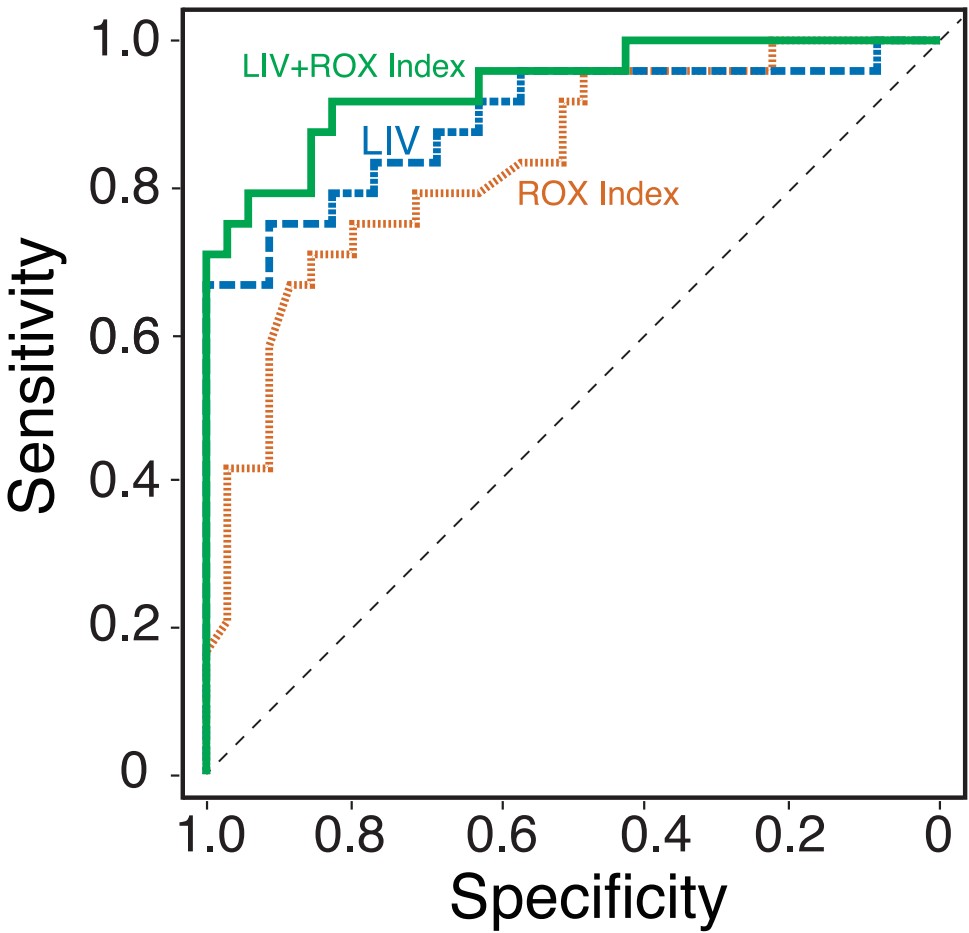

**Figure 5 ROC curves.** ROC curve results for the ROX index and LIV (AUC: 0.94, 95% CI [0.89–0.99], sensitivity: 0.88, specificity: 0.832) compared with the ROX index alone (AUC: 0.83, 95% CI [0.75–0.92], sensitivity: 0.79, specificity: 0.77) and LIV alone (AUC: 0.89, 95% CI [0.82–0.96], sensitivity: 0.79, specificity: 0.77). AUC, area under the curve; CI, confidence interval; LIV, lung infiltration volume; MV, mechanical ventilation; ROC, receiver operating characteristic; ROX index, ratio of oxygen saturation index.

(Table. S4). Patients above the SCORE are more likely to require MV, even if the ROX index is ≥ 6.1.

Sensitivity, specificity, accuracy, DOR, and the areas under the ROC curves were compared for three decision boundaries: ROX index alone, LIV alone, and ROX index and LIV combined (Fig. 5 and Table 3). As a result, in all binomial classification confidence indices, the decision boundary by the combination of ROX index and LIV showed better values (AUC: 0.94, sensitivity: 0.79, specificity: 0.91, accuracy: 0.86, DOR: 41) compared with values obtained from the ROX index alone (AUC: 0.83, sensitivity: 0.75, specificity: 0.80, accuracy: 0.78, DOR: 12) and LIV alone (AUC: 0.89, sensitivity: 0.75, specificity: 0.89, accuracy: 0.83, DOR: 23).

**Table 3  Indices for organ damage in HFNC and MV groups.**

| Cut-off parameters | Sensitivity | Specificity | Accuracy | PLR | NLR | DOR | AUC (95% CI) |
|---|---|---|---|---|---|---|---|
| ROX index | 0.75 | 0.80 | 0.78 | 3.75 | 0.31 | 12 | 0.83 (0.73–0.94) |
| LIV | 0.75 | 0.89 | 0.83 | 6.56 | 0.28 | 23 | 0.89 (0.80–0.98) |
| ROX index and LIV | 0.79 | 0.91 | 0.86 | 9.24 | 0.23 | 41 | 0.94 (0.88–0.99) |

**Notes.**

AUC, area under the curve; CI, confidence interval; DOR, diagnostic odds ratio; HFNC, high-flow nasal cannula; LIV, lung infiltration volume; MV, mechanical ventilation; NRL, negative likelihood ratio; PRL, positive likelihood ratio; ROX index, ratio of oxygen saturation index.

These findings suggest, in terms of the accuracy rate, that classification by the MLRA SCORE cut-off line was better than that by the cut-off of the ROX index alone or LIV alone.

This MLRA analysis excluded gender, age, and BMI from the main factors influencing the need for MV, as stated. However, there are many reports in which these factors are involved in the aggravation of COVID-19. Therefore, we confirmed whether these factors affected the need for MV and whether they affected the grouping according to the three cut-off lines. The results showed that only the number of cases with BMI >25 and BMI ≤ 25 showed statistically significant uneven distribution between HFNC-positive in mild cases and MV-positive in severe cases classified by the cut-off line of LIV alone, as shown in Table S5 . Gender and age (≥ 65 years, <65 years) did not significantly affect the use of HFNC and MV in our patient cohort.

# DISCUSSION

For AHRF caused by COVID-19, physicians are faced with the choice of respiratory therapy, such as HFNC or MV. If the patient's respiratory status can be managed with HFNC, it is essential that physicians do not carelessly introduce MV treatment, which places a heavy burden on both the patient and medical staff. Therefore, in the choice of HFNC or MV management in the treatment of COVID-19, the ROX index was proposed as a clinical indicator (*Roca et al., 2019*; *Roca et al., 2016b*). However, the reported cut-off value of the ROX index ranges widely from 2.7 to 5.9 (*Junhai et al., 2022*). In an early study, the cut-off value 6–12 h after receiving HFNC was reported as 4.88, with a 95% CI [4.2–5.4] (*Roca et al., 2019*; *Roca et al., 2016b*). A meta-analysis of COVID-19 patients with AHRF suggested that the ROX index is an excellent indicator for the prediction of HFNC failure although the cut-off value of the index varied from 2.7 to 5.99 (*Prakash et al., 2021*). Other recent meta-analyses demonstrated that a high chance of successful therapy is expected if a patient's ROX index is >5.4, and that patients are at an increased risk of HFNC failure and should be considered to require escalation of respiratory support if the ROX index is <4.2 (*Zhou et al., 2022*). Additionally, a cut-off value of the ROX index of >5 indicates expected successful weaning from HFNC (*Junhai et al., 2022*).

In the ex-post analysis of the ROX index cut-off value in our case, a slightly higher value of 6.1 was detected, probably because the clinicians in charge decided to transition patients from HFNC to MV when the severity of the lung injury on CT images was high even though the ROX index exceeded 5. Seven patients (20% of 35 HFNC patients) whose ROX index

values were ≤ 6.0 successfully recovered with HFNC alone, and seven patients (29.1% of 24 MV patients) whose ROX index values were >6.0 were treated with MV. Unfortunately, two of the patients with ROX values >6.0 of died. These patients had significantly high lung injury severity. Therefore, choosing to initiate MV based solely on the ROX index may create a high healthcare burden given the presence of COVID-19 patients with a variety of pathologies, and more complex criteria may be required to achieve higher sensitivity and specificity.

As an additional clinical parameter to support the ROX index in clinical judgment, lung injury severity assessment from chest CT images (LIV), as proposed in this case series, is one option. Recently, attempts to evaluate the severity of lung injury by scoring CT images of AHRF due to COVID-19 have also been reported. As one example, the total severity score (TSS) score is a scoring system in which the ratio of the volume of infiltrative shadows in the lung is scored on a 5-point scale for each of the five lobes of the lung, and a total score ranging from 0 to 20 is calculated (*Kucuk et al., 2022*; *Li et al., 2020*; *Tharwat et al., 2022*). High TTS scores indicate more severe disease. However, the TSS score requires a radiologist's evaluation of CT images, including anatomical assessment of lung lobes. Although the computer-analyzed LIV that we used is a straightforward method to evaluate lung injury, in this study, we were unable to confirm a correlation with a specific diagnosis of lung injury. Optimization of software technology related to the evaluation of a correlation with actual lung injury is a future issue. Moreover, we believe that there is room for further examination of the composite judgment criteria proposed by other researchers. For example, the prediction of the ROX index may be improved by combining the index with different parameters, such as the Sequential Organ Failure Assessment score (*Mellado-Artigas et al., 2021b*) and heart rate (*Goh et al., 2020*). HACOR, which is a prediction index for non-invasive MV failure (*Duan et al., 2017*), is an acronym for heart rate, acidosis, state of consciousness, oxygenation, and respiratory rate, and this index was reported to work successfully as a prediction index for MV in HFNC patients (*Valencia et al., 2021*).

In the present study, based on MLRA, the severity of lung injury calculated from chest CT images was added to the patient evaluation, with the ROX index. Patients with AHRF from COVID-19 pneumonia present with highly-variable pathophysiological characteristics, such as respiratory mechanics and responses to the prone position and recruitment maneuvers, despite a similar degree of hypoxemia (*Rossi et al., 2022*). Therefore, we suspected that some critically ill COVID-19 patients might require MV management even if their ROX index was higher than the reported cut-off value.

Recently, the ROX index has been studied for predicting hospitalization and mortality in patients with a diagnosis of COVID-19 in the emergency department and at hospital admission (*Gianstefani et al., 2021*; *Mukhtar et al., 2021*). A simple and low-cost clinical index, such as the ROX index, will be actively used and evaluated for prognosis estimation related to clinical judgment and triage for AHRF other than that caused by COVID-19 in the future. In addition to quick and easy to use indices, such as the ROX index, that support physicians' decisions, as shown in this study, the importance of additional tests and evaluation methods will likely increase in various medical situations.

This study evaluated the relationship between physicians' decisions about respiratory therapy selection and indices supporting the decision but not the outcome of the therapeutic choice. Additionally, our data were derived from a small number of patients at a single institution, and it is difficult to compare our data with other big data. Therefore, we do not propose a definitive cut-off value of the ROX index to improve clinical outcomes. Based on our experience in this case series, we suggest that it may be possible to construct a complex diagnostic criterion that will lead to better clinical judgment for respiratory therapy selection.

## CONCLUSIONS

Our study demonstrates that, by evaluating the pathophysiology of COVID-19 respiratory distress by adding the extent of the anatomical severity of pneumonia *via* chest CT to the ROX index, supportive guidance for physicians' decisions regarding respiratory management, either HFNC or MV, can be achieved for severely ill COVID-19 patients. This was a single-center retrospective study, and a prospective multicenter study of statistically processed predictive probabilities is needed.

## ACKNOWLEDGEMENTS

Concerning the basis of this clinical study, we would like to express our thanks to the clinicians in the intensive care unit (Prof. Satoru Hashimoto and attending doctors), infectious disease department, emergency department (Prof. Bon Ohta and attending doctors), general medical department, and internal medicine, and the ward nurses and laboratory technicians at the Hospital of the KPUM for their efforts in managing COVID-19 patients. We thank Hugh McGonigle, and Jane Charbonneau, DVM, from Edanz, for editing a draft of the manuscript.

### Funding
The authors received no funding for this work.

### Competing Interests
The authors declare there are no competing interests.

### Author Contributions
- Kazuki Sudo conceived and designed the experiments, performed the experiments, analyzed the data, prepared figures and/or tables, authored or reviewed drafts of the article, patient care, and approved the final draft.
- Teiji Sawa conceived and designed the experiments, performed the experiments, analyzed the data, prepared figures and/or tables, authored or reviewed drafts of the article, and approved the final draft.
- Kohsuke Kushimoto performed the experiments, authored or reviewed drafts of the article, patient care, and approved the final draft.

- Ryogo Yoshii performed the experiments, authored or reviewed drafts of the article, patient care, and approved the final draft.
- Kento Yuasa performed the experiments, authored or reviewed drafts of the article, patient care, and approved the final draft.
- Keita Inoue performed the experiments, authored or reviewed drafts of the article, patient care, and approved the final draft.
- Mao Kinoshita performed the experiments, authored or reviewed drafts of the article, patient care, and approved the final draft.
- Masaki Yamasaki performed the experiments, authored or reviewed drafts of the article, patient care, and approved the final draft.
- Kunihiko Kooguchi analyzed the data, prepared figures and/or tables, authored or reviewed drafts of the article, and approved the final draft.

## Human Ethics

The following information was supplied relating to ethical approvals (i.e., approving body and any reference numbers):

The Institutional Review Board of Kyoto Prefectural University of Medicine.

## Data Availability

The raw data of all 59 patients is available in the Supplementary File.

## Supplemental Information

Supplemental information for this article can be found online at http://dx.doi.org/10.7717/peerj.15174#supplemental-information.

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
