# Peer review of "Choice of respiratory therapy for COVID-19 patients with acute hypoxemic respiratory failure: a retrospective case series study"

_PeerJ, doi:10.7717/peerj.15174_

## Round 0.1 · original submission · Major Revisions

Specific comments are provided by three reviewers and I concur with the majority of those comments. Please address all of them properly.

Reviewer 1 ·

Basic reporting

This study retrospectively investigated the decision boundary of mechanical ventilation (MV) use by physicians for COVID-19 treatment using the ratio of oxygen saturation (ROX) index and lung infiltration volume (LIV). Although the sample size is small (n=59), the evaluation of the physician’s decisions using objective indices is in the readers' interest. However, the objective of this study is not well explained in the text. The authors mentioned “to devise a better criterion … (line 19, or 90)”; but a better criterion for what should be explained. In addition, “the accuracy of clinical decisions (line 42)” itself was not evaluated in this study since the outcome used in this study was not clinical outcomes (alive/dead or cured/not cured) but rather, the outcome was the physician’s empirical decision to use MV or not (i.e., the authors did not investigate whether the decision to achieve a good clinical outcome was accurate or not). So, as long as I understand, the objective of this study is to find objective indices to achieve empirical physician’s decisions of MV use (that contributes to shortening the time lag to use MV), which gives insights to shorten the delay from HFNC use to MV.

1. Line 19 and 90, a better criterion for what? A better criterion for MV initiation may be not enough because readers could misunderstand it as a criterion to achieve a good clinical outcome, but as I mentioned earlier, this study is to obtain a criterion to achieve the empirical physician’s decisions.
2. Line 42, the accuracy of clinical decisions: the same logic as line 19 applies here as well. The outcome variable of the analysis in this study is MV use or not by physician’s decision, but not a clinical outcome (dead/alive) (i.e., this study did not investigate the accuracy of clinical decisions, but evaluated a criterion that accurately judges the physician’s decision. Whether the clinical judgement was accurate or no is another problem).
3. Table S4: Sample sizes in the table are not the study population defined in the method. They should be explained in the text. All raw data was not provided for that sample size. Was the table not updated to the latest analysis?
4. Line 225: “38 patients who underwent MV” is different from that of the management group (24 patients for MV). The sample size used for each analysis (and why they were different) should be explained in the method. Also, please provide raw data for all 38 patients who underwent MV.
5. Figure 4: Please add legends for the ROX of 6.1 and LIV of 35.5.

Experimental design

The results are relatively well-written, but the descriptions of the analyses suddenly appeared in the result section (e.g., decision boundary using logistic regression) without the detail of it in the method section. What was done in this study should be clearly described in the methods.
Another important issue that should be mentioned in the manuscript is the number of MV available to use in the facility during the study period because the lack of MV should influence the physician’s decisions.

6. Line 40, the area under the curve of what? The authors did not explain ROC in the abstract.
7. Lines 67-71: the simple interpretation of ROX may be added such as “the lower, the severer”.
8. Lines 195-198: the methodology for Fig 3 is explained in the result section shortly, but it should be included in the method section with the objective of this plot (e.g., the meaning of the crossing point in Fig 3, and why it can be used as the cut-off value as described in lines 205-207).
9. Lines 213-219: this information should be included in the MLRA section in the method.
10. Lines 226-228: what the boundary score is and how they (equations) are derived were not explained in the method. Also, how the cut-off values (6.1 for ROX and 35.5 for LIV) are calculated was unclear with this result section only.
11. Lines 241-244: the same issue as the above (lines 226-228). The method for decision boundary using logistic regression is lacking. What MLRA SCORE is and how the cut-off line (MLRA SCORE LIV = 4.26 × (ROX index) + 7.89) was obtained should be explained in the method with references.
12. Table S4: the cut-off values and line are different from the main text. Was the table not updated to the latest analysis?
13. Lines 250-259: The ROC analysis was not described in the method. A simple explanation is needed in the method including the purpose of this analysis.

Validity of the findings

14. Line 320, “more appropriate guidance”: As I mentioned earlier (in basic reporting), this study evaluated the choice of respiratory management by physicians objectively using indices rather it investigated the appropriate use for it since the outcome of the analysis is MV-use/no-MV-use, which depends on physician’s decision, but not dead/alive. Whether physicians’ decision is appropriate or not is another problem although this study gives valuable insights into shortening the time lag to use MV.

Additional comments

15. Line 135, MV group: this term appeared for the first time here in the main text. Please define the term in the first paragraph of the methods.
16. Lines 198 and 199, rate: should be proportion (or percentage)
17. Lines 246-248: unclear expressions. Did the authors mean to tell the sensitivity and specificity of the ROX index are better compared to the other index (LIV and ROX+LIV)? Why did the authors mention about ROX index only to explain Table 3?
18. Lines 250-259: the complete repetition from Table 3 should be avoided.
19. Lines 265-267: how did the authors judge they were uniformly distributed? (by the author's view? Or did authors conduct a statistical test?)

·

Basic reporting

The article is well-written
However, the authors can improve the abstract. Instead of writing the calculations in the background section, they can instead start with a description of the problem statement, and then build up on the topic

Experimental design

At multiple places through out the manuscript (like table 3 of supplementary, and Line 156 of main text), numbers are described. Descriptive statistics should carry a mention of the type of data expression: mean (SD), or median (IQR) or something else.

In the materials and methods section, the authors can specify the study objective or aim in a subsection. It is not clearly mentioned now.

Validity of the findings

There are several different CT severity assessment scores currently in use. The authors can compare and discuss their findings (with LIV) with these other scores and give an explanation.

Research is ongoing regarding the importance of the variation in ROX in prognosis of COVID-19. The authors can mention this for the comprehensiveness of the discussion since this is highly relevant to the topic under discussion.

It is not clear how these results provide this conclusion. The authors should elaborate on this. Lines 207-210: “… This
Manuscript to be reviewed
208 finding suggests that adding LIV evaluation to the treatment policy decision may better
209 contribute to reducing false positives and false negatives compared with setting a more stringent
210 ROX index cut-off value...”

The authors can mention what inference do we draw from the kernel density estimation plots here.

Reviewer 3 ·

Basic reporting

1. Since the study discussed the “choice of respiratory therapy”, did the authors have any recommendation about which therapy should be provided given certain conditions?
2. “Informed consent was obtained from all subjects and/or their legal guardian(s), …” Were non-adults also included in the study? If yes, did the authors apply any adjustments? Please clarify.

Experimental design

No comments.

Validity of the findings

1. In line 226 “the boundary score (SCORE) for classifying patients selected for MV was calculated as SCORE = 21.50.0.81 × [ROX index] + 0.19 × [LIV].” How were the coefficients obtained? Same question for “the cut-off line by MLRA SCORE LIV = 4.26 × (ROX index) + 7.89 …”.
2. How was the validation performed? Please provide some information.

---

## Round 0.2 · Minor Revisions

Please address remaining specific comments. It's likely that this will be the final round, and thus, please submit the revised version that you can live with.

Reviewer 1 ·

Basic reporting

The manuscript has significantly improved. I left some minor comments to be addressed.

1. Lines 159-200:
The contents are understandable, but the explanation is not clear maybe with too much information. What the authors did was simply “plotting the percentages of MV cases with a ROX index > ROX cut-off (true positive ratio, or sensitivity) curve and the percentages of HFNC cases with a ROX index < ROX cut-off (true negative ratio, or specificity) curve vs cut-off values to identify a threshold for MV use by physicians”. Are the descriptions of true/false positive/negative (L161-165, L169-183, and Lines 191-200) necessary?

2. Lines 165-168:
The authors defined the crossing point as the MV use cut-off point. Having a reference for this such as the below may be good.
Greiner M, Sohr D, Göbel P. A modified ROC analysis for the selection of cut-off values and the definition of intermediate results of serodiagnostic tests. J Immunol Methods (1995) 185:123–132. doi: 10.1016/0022-1759(95)00121-P
Yoshida T, Kimura M, Yamada Y, Yokoyama K, Ishihara T, Yoshinaka Y, Itoi A, Watanabe Y, Kida N, Nomura T, et al. Fitness Age Score and the Risk of Long-Term Care Insurance Certification—The Kyoto-Kameoka Longitudinal Study. Open J Epidemiol (2017) 7:190–200. doi: 10.4236/OJEPI.2017.72016

3. Line 300, rate of MV: percentage (or proportion) of MV
4. Line 325 Fig.S1: should it be Fig. S2?

Experimental design

no comment

Validity of the findings

no comment

Additional comments

no comment

Reviewer 3 ·

Basic reporting

No comment.

Experimental design

No comment.

Validity of the findings

No comment.

---

## Round 0.3 · accepted · Accept

The authors are to be congratulated on their successful revisions.